# Systemic Sclerosis and Vaccinations: A Register-Based Cohort Study about Seasonal Influenza and *Streptococcus pneumoniae* Vaccination Rate and Uptake from Liguria Regional Center, Northwest Italy

**DOI:** 10.3390/vaccines8020204

**Published:** 2020-04-28

**Authors:** Giuseppe Murdaca, Giovanni Noberasco, Alberto Battaglini, Chiara Vassallo, Francesca Giusti, Monica Greco, Chiara Schiavi, Laura Sticchi, Giancarlo Icardi, Andrea Orsi

**Affiliations:** 1Departments of Internal Medicine, Scleroderma Unit, Clinical Immunology Unit, University of Genova, 16132 Genova, Italy; chiaravsl@gmail.com (C.V.); fra333.90@gmail.com (F.G.); monicagreco89@gmail.com (M.G.); chiaschiavi@libero.it (C.S.); 2Department of Health Sciences, Vaccines and Clinical Trials Unit, University of Genova, 16132 Genova, Italy; noberasco.giovanni@gmail.com (G.N.); Battaglini.Alb@gmail.com (A.B.); laura.sticchi@unige.it (L.S.); icardi@unige.it (G.I.); andrea.orsi@unige.it (A.O.); 3Hygiene Unit, “Ospedale Policlinico San Martino IRCCS”, 16132 Genova, Italy

**Keywords:** systemic sclerosis, vaccines, flu, influenza, pneumonia, *Streptococcus pneumoniae*

## Abstract

Systemic sclerosis (SSc) is the connective tissue disease with the highest mortality and patients with chronic inflammatory immune-mediated diseases are at high risk of acquiring infections as they are often treated with immunosuppressive or biological drugs. This study, conducted among the patients followed by our clinical immunology, part of the Internal Medicine Department in the Ospedale Policlinico San Martino, Genoa, northwest Italy, has set itself the primary objective of analyzing the vaccine uptake and the vaccination coverage against both seasonal influenza and **S. pneumoniae** in a cohort of patients with SSc. We evaluated the influenza and pneumococcal vaccination rate among various subgroups of patients and the source of the recommendation for vaccination. We evaluated the vaccination rate changes between the two years considered in our study. We also calculated a binomial logistic regression between vaccination acceptance and clinical and demographics characteristics of the patients to evaluate the adjusted odds ratio (OR) of each factor on vaccination. The vaccination coverage that resulted was significantly higher than in other similar studies. Age over 65 years old, interstitial lung disease, and ongoing immunosuppressive therapy were significantly related with acceptance to both vaccinations using univariate analyses, but the multivariate logistic regression found a significant correlation only with the age and therapy factors.

## 1. Introduction

Systemic sclerosis (SSc) is a rare connective tissue disease of unknown etiology characterized by chronic inflammation and fibrosis of the skin, vascular abnormalities, and variable involvement of organs including kidneys, gastrointestinal tract, heart, and lungs [1,2]. SSc is classified into two main subtypes: limited systemic sclerosis (lSSc) and diffuse systemic sclerosis (dSSc). Given the heterogeneity of clinical symptoms and signs, the American College of Rheumatology (ACR) and the European League Against Rheumatism (EULAR) recently developed new classification criteria [3], which would lead to earlier diagnosis and also incorporate the autoantibodies that are commonly used for diagnostic purposes. The prevalence of SSc ranges from 7 to 700 cases per million owing to a substantial variation across geographic regions and ethnic differences. However, SSc seems to be consistently more frequent in the USA and Australia as compared with Europe and Japan [4,5]. SSc is the connective tissue disease with the highest mortality. Systematic reviews of cohort studies and meta-analyses showed a pooled standardized mortality ratio of 2.3–3.5 in SSc, with a cumulative survival from diagnosis of 75% at 5 years and 62.5% at 10 years [6]. Interstitial lung disease (ILD) and pulmonary arterial hypertension (PAH) cause more than half of all SSc-related deaths [7]. Indeed, patients with diffuse cutaneous SSc are at a high risk of developing severe respiratory complications such as to induce hospitalization in an intensive care unit (ICU). Notably, the time from diagnosis to ICU admission was 78 months and the main reason for ICU admission was acute respiratory failure [8]. However, patients with SSc live longer and cardiac deaths are increasing extremely favored by endothelial dysfunction and vascular abnormalities [9,10]. SSc shows a complex etiology that still appears to be not fully clarified even if both environmental and genetic factors seem to influence the onset and outcome of the disease [11]. Furthermore, SSc is known to be associated with abnormal T-cell immunoregulation. In particular, the function of CD8+ suppressor cells is impaired in patients with systemic lupus erythematosus (SLE) in relapse and in patients with SSc with disease progression, suggesting the involvement of CD8+ suppressor cells in the pathogenesis of chronic inflammatory immune-mediated diseases [12]. However, patients with chronic inflammatory immune-mediated diseases are at a high risk of acquiring even opportunistic infections as they are often treated with immunosuppressive or biological drugs [13,14,15,16,17]. Patients with chronic inflammatory immune-mediated diseases are at higher risk for influenza and *Streptococcus pneumoniae* (*S. pneumoniae*) infections [18,19,20]. However, influenza infection can be self-limiting, but also lead to severe complications like pneumonia or organ failure, depending on factors such as the patient’s age, comorbidities, and immunosuppressive drugs [19]. The risk of getting influenza infection and developing severe complications is higher in patients with chronic inflammatory immune-mediated diseases than in the average population. Indeed, a large cohort study showed that the incidence of influenza complications was 2.75-fold higher in patients with rheumatoid arthritis [20]. The current EULAR guideline about vaccination for patients with rheumatic diseases strongly recommends vaccination against seasonal influenza [21]. *S. pneumoniae* is the most important cause of pneumonia, meningitis, and bacterial sepsis worldwide [22], particularly in young children and elderly subjects who are more susceptible to infection [23,24] and when receiving aggressive long-term immunosuppressive therapy [25,26,27]. Some *S. pneumoniae* serotypes are associated with higher mortality, whereas other serotypes show lower case-fatality rates [28]. In the case of invasive disease, mortality rates vary from 5% to 35% [29]. Considering the importance that vaccines hold in patients with immunity deficiency in preventing infectious diseases, important causes of mortality and morbidity, this study has set itself the primary objective of analyzing the vaccination coverage against both seasonal influenza and *S. pneumoniae* in a cohort of patients with SSc, and investigating demographic and clinical factors significantly related to vaccine acceptance and uptake.

## 2. Materials and Methods

The study was conducted among the patients followed by our clinical immunology, part of the Internal Medicine Department in the Ospedale Policlinico San Martino, Genoa, northwest Italy. This is the reference regional center for scleroderma diagnosis and treatment. Starting from September 2018, all patients aged above 18 years old were proposed an informed consent document (which was kept in their medical records) about this study participation; after that, information was collected. Information collected was on sex, age, influenza vaccination status, pneumococcal vaccination status, immunosuppressive therapy, drugs used, presence of comorbidities (interstitial lung disease, cardiovascular disease, diabetes, chronic kidney failure, presence of hematological or oncological malignancies), smoke habit, and source of the vaccination recommendation. We referred all patients to our department of hygiene for an evaluation, and the patients who were not vaccinated before and who consented to the vaccination protocol received the seasonal flu vaccine, the 13-valent conjugate vaccine, and the re-challenge with the 23-valent polysaccharide vaccine. The data were combined with our data regarding the patient’s health and disease progression, organ involvement, significant comorbidities, and concurrent therapies, with particular regard to the use of immunosuppressive agents. This survey was carried out on two consecutive years in order to confront the vaccination rates and evaluate their characteristics. A statistical analysis of the data obtained up to this point was performed. We evaluated the influenza and pneumococcal vaccination rate and acceptance among various subgroups of patients such as patients with comorbidities, age over 65, immunosuppressive therapy, and source of the recommendation for vaccination. We evaluated the vaccination rate changes between the two years considered in our study. Quantitative variables are described with means, medians, and interquartile ranges (IQRs); qualitative variables are described with proportions and percentages. The correlation between vaccine uptake and other factors was identified with Fisher’s exact test and Yates’s chi-squared test. IBM SPSS Statistics version 25 (manufactured by IBM in Armonk, NY, USA) was used to calculate a binomial logistic regression between vaccination and clinical and demographics characteristics of the patients in order to evaluate the adjuster odds ratio (OR) of each factor on vaccination.

## 3. Results

### 3.1. Clinical and Demographic Characteristics

Seventy-two patients were recruited in the first year and 91 patients in the second year. One patient was excluded from the study owing to death unrelated to the study (myocardial infarction and subsequent cardiac failure, without mention of infective complications).

The majority (85.71%) of the patients had significant comorbidities, in particular, 58.24% of the patients had some form of interstitial lung disease (ILD), 53.85% of the patients had heart-related comorbidities, 18.64% of the patients had a chronic kidney disease (CKD) stage II higher grade of or kidney failure, and 4.4% of the patients were in active treatment or strict follow-up for a concurrent neoplastic disease. About one-third (34.07%) of the patients were active cigarette smokers, while 40.66% of the patients received immunosuppressive therapy, with the most common regimen being with mofetil-mycophenolate (27.47%) and methotrexate (8.79%).

Clinical and demographic characteristics of our patients are described in Table 1.

### 3.2. Vaccination Rate

We observed significant changes between vaccination rates in the two seasons evaluated in this study.

Vaccination rate changes are described in Table 2 and Table 3.

### 3.3. Vaccinations Uptake

According to Italian guidelines on immunization [30], only 7 out of 91 patients (7.7%) had no indications for vaccination, even if systemic sclerosis is not explicitly mentioned as an indication.

Forty-three out of 91 patients were aged over 65 years, which is an indication for both flu and pneumococcal vaccination. Out of the remaining 48 patients, 39 had at least one comorbidity, fitting with the guideline recommendations for immunization.

Among the nine patients without any comorbidity, two were under immunosuppressive therapy with methotrexate.

In the group of patients with an actual recommendation for both vaccines, we found coverage of 78.6% and 75.0% for pneumococcal and flu vaccination, respectively.

Sixty-seven patients (73.6%) were recommended the pneumococcal vaccination; the recommendation came from the immunologist for 85.1% of the patients, from the general physician for 11.9%, and from another specialized doctor in 3% of the patients.

One hundred percent of these patients accepted the vaccination.

Among the sixty patients (65.9%) who received a recommendation for the flu vaccination, the sources were the immunologist and the general physician for 88.3% and 11.7% of the patients, respectively.

One hundred percent of these patients accepted the vaccination.

The recommendation from the physician is significantly related with the vaccine uptake (*p* < 0.0001 for both vaccines).

A significant correlation between both vaccinations and ongoing immunosuppressive therapy (*p* = 0.008 for *S. pneumoniae* and *p* = 0.01 for flu) was found with Fisher’s exact test and Yates’s chi-squared test. Significant correlations were also found between both vaccinations and the presence of interstitial lung disease (ILD) (*p* = 0.01 for *S. pneumoniae* and *p* = 0.004 for flu, adjusted with Yates) and age over 65 years (*p* = 0.019 for *S. pneumoniae*, Yates adjusted, and *p* = 0.0002 for flu, Yates adjusted).

Using IBM SPSS Statistics version 25, we ran a binomial logistic regression between flu and pneumococcal vaccination and patients’ characteristics that may be related to vaccination.

The characteristics considered were sex, age (under/over 65 years old), smoking habit, ongoing immunosuppressive therapy, ILD, pulmonary arterial hypertension (PAH), chronic obstructive pulmonary disease (COPD), cardiovascular disease (CV), diabetes (DM), chronic kidney failure (CKD), presence of hematological or oncological malignancies, and number of comorbidities (0–1 vs. 2 or more).

Table 4 and Table 5 present the raw and adjusted odds ratios of each factor on flu and pneumococcal vaccine uptake, respectively. Age over 65 years and ongoing immunosuppressive therapy were confirmed as characteristics strongly linked with both flu vaccination (OR = 31.681 and OR = 31.979, respectively) and pneumococcal vaccination (OR = 6.098 and OR = 11.236, respectively) acceptance. The presence of cardiovascular comorbidity showed a significant adjusted odds ratio with pneumococcal vaccination (OR = 8.184), but not with flu vaccination. The adjusted odds ratios were not measurable between both vaccinations and COPD and diabetes.

## 4. Discussion

Seasonal influenza and pneumococcal infections play an important role among respiratory infections in terms of incidence, morbidity, and mortality, and studies showed an even higher incidence and mortality in patients with autoimmune diseases like systemic sclerosis [4,6,13,21,23], considering the important weight on health care systems SSc is estimated to have [31]. The vaccinations considered in this study showed a high level of safety and efficacy in several aforementioned studies [19,23,25,26,27], and they should be recommended from both general physicians and specialized doctors; alongside this work, other studies showed that patients who received a direct recommendation from a physician are significantly more likely to be vaccinated than the others [32]. Patients followed by specialist care are also more likely to be vaccinated. The aim of this study was to evaluate the vaccination coverage against influenza and *S. pneumoniae* in a cohort of patients with scleroderma. The vaccination coverage was significantly higher than in other similar studies [33,34]. We found coverage of 78.6% and 75% for pneumococcal and flu vaccination, respectively, among patients with actual recommendations, according to the goals set in the Italian Prevention Plan 2017–2020 [30].

A flu vaccination coverage of 75% was the minimum set goal for patients over 65 years old (coverage in our cohort was 88.6%) and for patients with other risk conditions determined by comorbidities (75.0% coverage in our cohort) or ongoing immunosuppressive therapy (83.8% coverage in our cohort). Influenza vaccine coverages are yet far from the 95% optimal goal set in the Italian Prevention Plan, but they are much higher than the Italian overall (2017–2018: 58.3% vs. 15.3%; 2018–2019: 69.2% vs. 15.8%) and over 65 years old population coverage (2017–2018: 75.7% vs. 52.7%; 2018–2019: 88.6% vs. 53.1%) [35].

The Italian Prevention Plan set partial goals for each year regarding the pneumococcal vaccination coverage in patients older than 65 years old.

The first year evaluated in our study did not reach its objective, but at the end of the study, we managed to reach a coverage of 86.3% in patients older than 65 years old, much higher than the set goal of 75%, which was nearly reached even by the coverage of all the patients (74.7%).

Even if scleroderma does not represent a specific recommendation for immunization practice against flu and *S. pneumoniae*, we found that 92.3% of our patients had an actual recommendation such as age older than 65 years old, specific high-risk comorbidity, or ongoing immunosuppressive therapy.

Age over 65 years old, interstitial lung disease, and ongoing immunosuppressive therapy were significantly related to both vaccinations using univariate analyses, but the multivariate logistic regression found a significant correlation only with the age and therapy factors.

The positive correlation between both influenza and pneumococcal vaccination and vaccine adherence and older age (>65 years old) has already been highlighted in similar studies, which also found significantly higher vaccine coverage in patients undergoing immunosuppressive therapy such as oral glucocorticoids, methotrexate, leflunomide, and other disease-modifying antirheumatic drugs (DMARDs). Among different therapy groups, a stronger correlation between vaccine uptake and biological DMARDs treatment compared with conventional treatment was found [32].

A significant correlation between pneumococcal vaccination and the presence of cardiovascular disease was also found; severe chronic comorbidities are often associated with higher vaccine adherence (both influenza and pneumococcal vaccine) in similar studies [36].

The lack of adequate information and of a full and unambiguous indication is the most relevant possible cause of low vaccine coverage and the most referred cause of vaccine refusal [32].

The influence of the physician recommendations weighs significantly in the patient vaccine uptake; in order to enhance the strength of these recommendations, there is the clear need for guidelines indicating the role of general physicians and an explicit statement of systemic sclerosis as a condition where said vaccinations are strongly recommended in patients with systemic sclerosis, notwithstanding the severity of the disease, its therapy, or its associated conditions. Authors should discuss the results and how they can be interpreted in the perspective of previous studies and the working hypotheses. The findings and their implications should be discussed in the broadest context possible. Future research directions may also be highlighted.

## 5. Conclusions

In conclusion, we can say that the vaccine coverage was found to reach or almost reach a satisfying goal among patients with a risk factor aside systemic sclerosis, even though the overall coverage of these patients still needs interventions in order to improve.

## Figures and Tables

**Table 1 vaccines-08-00204-t001:** Clinical and demographic characteristics of systemic sclerosis (SSc) patients.

	Total	Patients Vaccinated against Flu	Patients Not Vaccinated against Flu	Patients Vaccinated against *S. pneumoniae*	Patients Not Vaccinated against *S. pneumoniae*
Number	91	63	28	68	23
Mean age	63.5	67.4	54.7	65.9	56.4
Median age	64	70	51.5	69	51
IQR ^1^	54–74.5	60–77.5	47.75–60.25	57.75–76	47.5–64.5
Over 65 years old	48.4%	61.9%	17.9%	55.9%	26.0%
Females	82.4%	81.0%	85.7%	82.4%	82.6%
Comorbidities	85.7%	93.7%	67.9%	91.2%	69.6%
ILD ^2^	58.2%	65.1%	42.9%	63.2%	43.5%
Cardiovascular	53.9%	58.7%	42.9%	57.4%	43.5%
Neoplastic	4.4%	4.8%	3.6%	4.4%	4.4%
CKD ^3^	18.7%	19.0%	17.9%	19.1%	17.4%
Diabetes	4.4%	6.4%	0.0%	5.9%	0.0%
Smokers	34.1%	31.8%	39.3%	32.4%	39.1%
Ongoing therapy	40.7%	49.2%	21.4%	48.5%	17.4%
MTX ^4^	8.8%	11.1%	3.6%	10.3%	4.4%
MMF ^5^	27.5%	33.3%	14.3%	32.4%	13.0%

^1^ IQR: interquartile range; ^2^ ILD: interstitial lung disease; ^3^ CKD: chronic kidney disease; ^4^ MTX: methotrexate; ^5^ MMF: mycophenolate mofetil.

**Table 2 vaccines-08-00204-t002:** Changes in flu vaccination coverage between the 2017–2018 and 2018–2019 seasons.

	2017–2018	2018–2019	*p*
Total (*n*)	58.3% (43)	69.2% (63)	0.206
Over 65 years old	75.7%	88.6%	0.003
Males	54.6%	75.0%	0.268
Females	59.0%	81.0%	0.004
Comorbidities	60.4%	75.0%	0.117
ILD ^1^	58.7%	77.4%	0.050
Cardiovascular	76.9%	75.5%	0.944
Ongoing therapy	34.9%	83.8%	<0.001
MTX ^2^	60.0%	87.5%	0.252
MMF ^3^	73.3%	84.0%	0.478

^1^ ILD: interstitial lung disease; ^2^ MTX: methotrexate; ^3^ MMF: mycophenolate mofetil.

**Table 3 vaccines-08-00204-t003:** Changes in *S. pneumoniae* vaccination coverage between the 2017–2018 and 2018–2019 seasons.

	2017–2018	2018–2019	*p*
Total (*n*)	23.6% (17)	74.7% (68)	<0.001
Over 65 years old	24.3%	86.3%	<0.001
Males	18.2%	75.0%	0.004
Females	24.6%%	82.4%	<0.001
Comorbidities	27.8%	78.8%	<0.001
ILD ^1^	30.4%	89.3%	<0.001
Cardiovascular	38.5%	85.7%	<0.001
Ongoing therapy	35.3%	89.2%	<0.001
MTX ^2^	40.0%	87.5%	0.070
MMF ^3^	26.7%	88.0%	<0.001

^1^ ILD: interstitial lung disease; ^2^ MTX: methotrexate; ^3^ MMF: mycophenolate mofetil.

**Table 4 vaccines-08-00204-t004:** Raw and adjusted odds ratios (ORs) on each factor evaluated for flu vaccination.

	Raw OR	Adjusted OR	95% Confidence Interval	*p*
Female sex	0.708	0.179	0.017–1.848	0.149
Age over 65 years old	8	31.681	4.889–205.287	<0.001
Immunosuppressive therapy	3.552	31.979	4.327–236.355	0.001
ILD ^1^	2.485	2.524	0.429–14.871	0.306
PAH ^2^	1.121	2.27	0.110–46.767	0.595
COPD ^3^	2.417	Not available	Not available	0.999
Cardiovascular	0.459	6.662	0.722–61.474	0.094
Diabetes	2.417	Not available	Not available	0.999
CKD ^4^	1.082	1.313	0.174–9.929	0.792
Neoplastic disease	1.35	8.754	0.193–396.689	0.265
Smoking	0.719	0.648	0.166–2.528	0.533
2 or more comorbidities	1.7	0.157	0.011–2.219	0.170

^1^ ILD: interstitial lung disease; ^2^ PAH: pulmonary arterial hypertension; ^3^ COPD: chronic obstructive pulmonary disease; ^4^ CKD: chronic kidney disease.

**Table 5 vaccines-08-00204-t005:** Raw and adjusted odds ratios on each factor evaluated for *S. pneumoniae* vaccination.

	Raw OR	Adjusted OR	95% Confidence Interval	*p*
Female sex	0.982	0.562	0.094–3.372	0.562
Age over 65 years old	3.81	6.098	1.431–25.985	0.014
Immunosuppressive therapy	4.479	11.236	42.384–52.965	0.002
ILD ^1^	2.236	2.955	0.552–15.817	0.205
PAH ^2^	0.833	1.271	0.117–13.806	0.844
COPD ^3^	1.846	Not available	Not available	0.999
Cardiovascular	1.857	8.184	1.101–60.859	0.040
Diabetes	1.846	Not available	Not available	0.999
CKD ^4^	1.123	2.056	0.295–14.353	0.467
Neoplastic disease	1.015	4.983	0.243–102.211	0.297
Smoking	0.744	0.766	0.208–2.819	0.699
2 or more comorbidities	1.3	0.096	0.008–1.178	0.067

^1^ ILD: interstitial lung disease; ^2^ PAH: pulmonary arterial hypertension; ^3^ COPD: chronic obstructive pulmonary disease; ^4^ CKD: chronic kidney disease.

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
