# Peer review of "Systemic Sclerosis and Vaccinations: A Register-Based Cohort Study about Seasonal Influenza and *Streptococcus pneumoniae* Vaccination Rate and Uptake from Liguria Regional Center, Northwest Italy"

_vaccines, 2020, doi:10.3390/vaccines8020204_

Round 1
Reviewer 1 Report
Murdaca et al presented their data about vaccinations in scleroderma patients. The data are original and the manuscript is well written.
Author Response
Thank you for your review.
Reviewer 2 Report
Review of Murdaca et al: Systemic sclerosis and vaccinations: a register-based cohort study about influenza and S. Pneumoniae vaccination rate from Liguria regional center, northwest Italy.
The current paper involves vaccination rate in Systemic sclerosis patients. The setup of the study is not clear. Patients that were not vaccinated for influenza and S. pneumonia were referred to the department of Hygiene for vaccination. Next the increase in vaccination rate is mentioned in tables 2 and 3 is mentioned. Correlations and OR’s are shown in tables 4 and 5.
Major comments.
- To me it is not clear what the authors are showing here. They refer patients to let them be vaccinated and then they calculate which characteristics are related to vaccination. To my opinion these characteristics are determined by the fact whether a patient wants to be vaccinated or not.
- The paper is poorly described, especially the part describing the results is very badly written.
- In table 1 too many digits after comma are shown.
- The authors do not show how the correlations and ORs are calculated. What is here correlated to what?
- In the abstract the authors mention significant differences but they do not state compared to what: Age over 65 years old, interstitial lung disease and ongoing immunosuppressive therapy resulted significantly related with both vaccinations using univariate analyses but the multivariate logistic regression found significant correlation only with the age and therapy factors.
- The references used are not up to date: the EULAR has published new guidelines last year.
- To my opinion is 25% selfcitation a bit too much!
Author Response
- The aim of this manuscript is to analyze vaccination coverage and to investigate demographic and clinical factors significantly related to vaccine acceptance and uptake.
- In order to achieve a clearer display the result section is now divided in three sub sections.
- The number of digits after the comma in table 1 was reduced.
- Significant correlation between vaccine uptake and other factors was identified with Fisher’s exact test and Yates’s chi squared test. IBM SPSS Statistics version 25 was used to calculate a binomial logistic regression between vaccination and clinical and demographics characteristics of the patients in order to evaluate the adjuster odds ratio (OR) of each factor on vaccination.
- Age over 65 years old, interstitial lung disease and ongoing immunosuppressive therapy resulted significantly related with both vaccinations acceptance and uptake. We compared the acceptance rates among sub-groups of patients based on these demographical and clinical characteristics.
- The references are now updated to the latest EULAR guidelines.
- The number of self-citacion is now reduced down to an 11% rate.
Reviewer 3 Report
The authors aimed to evaluate the vaccination coverage rate of seasonal influenza and S. pneumoniae by two successive cohorts in patients with systemic sclerosis subdivided into different categories observations in a single hospital. Some valuable conclusions are made. The following comments and suggestions are raised.
- The typing errors of the text need supervision by an experienced person to improve.
- The “comma” appears in the number should be replaced by “punctuation” in ttext and all tables.
- The medical term “influenza” in the title is had better changed to “seasonal influenza” for more accuracy.
- Ref. 15 is wrongly cited as Ref. 14 in P.2 regarding the CD8+ suppressor T cells.
- Table 2 and Table 3 are the comparison of flu and S. pneumoniae coverage rates between the two cohorts. It would be better to add the p values for the statistical significance.
- P.7: The paragraph of the “In conclusion” should be put in the end of “Discussion” for more clarity.
Author Response
- The typing errors of the text are supervised and corrected.
- The “comma” appearing in the number are replaced by “punctuation” in ttext and all tables.
- The medical term “influenza” in the title is now changed to “seasonal influenza” for more accuracy.
- References number 14 and 15 are now correct.
- p values for the statistical significance are now added in Table 2 and Table 3.
- The order of the Discussion section is now more clear.
Round 2
Reviewer 2 Report
The authors made appropriate revisions. They should discuss the data in tables 4 and 5 concerning COPD and Diabetes. It is not clear what these numbers show.
Author Response
The manuscript underwent an appropriate English revision.
It was impossible to measure the adjusted odds ratios for COPD and diabetes and it is now more clearly stated both in said tables and in the manuscript.
Reviewer 3 Report
No
Author Response
Thank you for your review.
Kind regards.